# All-dielectric scale invariant waveguide

Janderson R. Rodrigues[1], Utsav D. Dave[1], Aseema Mohanty [2], Xingchen Ji [3], Ipshita Datta [1], Shriddha Chaitanya[1], Euijae Shim[1], Ricardo Gutierrez-Jauregui [4], Vilson R. Almeida[5,6], Ana Asenjo-Garcia[4] & Michal Lipson [1] ✉

Total internal reflection (TIR) governs the guiding mechanisms of almost all dielectric waveguides and therefore constrains most of the light in the material with the highest refractive index. The few options available to access the properties of lower-index materials include designs that are either lossy, periodic, exhibit limited optical bandwidth or are restricted to subwavelength modal volumes. Here, we propose and demonstrate a guiding mechanism that leverages symmetry in multilayer dielectric waveguides as well as evanescent fields to strongly confine light in low-index materials. The proposed waveguide structures exhibit unusual light properties, such as uniform field distribution with a non-Gaussian spatial profile and scale invariance of the optical mode. This guiding mechanism is general and can be further extended to various optical structures, employed for different polarizations, and in different spectral regions. Therefore, our results can have huge implications for integrated photonics and related technologies.

Heterogeneous integration of photonics platforms is essential for the development of new classes of classical[1–4] and quantum applications[5]. The ability to use light to explore different physical properties, such as electrical gating, optical gain or loss, non-linearities, and thermal tuning, on the same chip, has the capability of revolutionizing photonics, by paving the way to technological breakthroughs and mass adoption[6–9]. However, the addition of new materials on the existing platforms, each one with its refractive index, reveals new challenges. One of these is related to the spatial distribution of light in waveguides, which is fundamentally limited to the material with the highest refractive index by total internal reflection (TIR). This critical challenge limits the amount of light-matter interaction with low-refractive-index materials, which in turn restricts their usage.

To date, guiding light in low-index materials has been achieved using Bragg reflectors[10], antiresonant structures[11–14], photonic crystals[15–17], slot-waveguides[18,19], plasmonic waveguides[20–23], and ALIG waveguides[24,25]. However, all of these solutions are either lossy, require periodicity (larger footprint), exhibit narrow bandwidth, are

polarization restricted[18,25], or are limited to subwavelength dimensions[18,20,26]. As an example, consider the slot waveguide that confines light in the low-index media by relying on the discontinuity of the electric field at the dielectric interfaces between the two media. The effect occurs only for a given polarization and when light is confined to subwavelength sizes which, besides imposing rigid constraints in the fabrication process, limits the amount of interaction[18].

In this work, we show a guiding mechanism that harnesses evanescent fields of the propagating modes in lower-index materials without importing the typical high losses associated with radiation modes. We explore the protection ensured by a mirror-symmetric waveguide structure in order to gain access to these previously inaccessible lossless regimes. This guiding mechanism leads to a uniform field distribution in the low-index material with a non-Gaussian profile and the scale invariance of the optical mode, which facilitates full control of the strength of the light-matter interaction process. Our design is robust to fabrication variations, is broadband, is polarization-independent, and most importantly, introduces no additional optical loss.

[1]Department of Electrical Engineering, Columbia University, New York, NY 10027, USA. [2]Department of Electrical and Computer Engineering, Tufts University, Medford, MA 02155, USA. [3]John Hopcroft Center for Computer Science, School of Electronic Information and Electrical Engineering, Shanghai Jiao Tong University, 200240 Shanghai, China. [4]Department of Physics, Columbia University, New York, NY 10027, USA. [5]Aeronautics Institute of Technology, Sao Jose dos Campos, SP 12228, Brazil. [6]Universidade Brasil, Sao Paulo, SP 08230, Brazil. ✉e-mail: ml3745@columbia.edu

# Results

## Guiding principle

The guiding principle can be understood from the planar waveguide structure sketched in Fig. 1a. We consider two higher-index-material slabs with equal thickness $t/2$ and refractive index $n_H$, separated by an intermediary-index-material slab of thickness $d$ and index $n_S$, and cladded by a lower-index material with index $n_L$, where $n_H>n_S>n_L$. For the TE (Transverse Electric) polarization, the electric field $E_x$ propagates along the $z$-direction inside of the middle slab ($|y| \leq d/2$) and has spatial dependence in the $y$-direction given by $E_x(y) = |E_0| \cosh(\gamma_S y)$, where $|E_0|$ is the electric field maximum amplitude and $\gamma_S$ is the internal field decay coefficient in the middle layer (see Supplementary Section 1 for the full description of the field distribution). The transverse wavevector in the middle layer, $k_S$, is related to the field decay through the relation $k_S = \pm i\gamma_S = \pm ik_0\sqrt{n_{eff}^2 - n_S^2}$, where $k_0$ is the wavevector magnitude, which is given by $k_0 = 2\pi/\lambda_0$, where $\lambda_0$ is the wavelength of light in vacuum. The optical mode is characterized by its effective refractive index $n_{eff}$, which is obtained from the dielectric boundary conditions and the energy conservation requirements. In the conventional regime, light is confined to the highest-index slab, $k_S$ is purely imaginary ($n_{eff}>n_S$) and the field distribution represents an evanescent field decay inside the middle region (see Fig. 1b). On the other hand, when $k_S$ is zero ($n_{eff} = n_S$) the solution is a uniform field throughout the middle slab (i.e., $E_x(y) = |E_0|$), regardless of its thickness. Furthermore, when $k_S$ is purely real ($n_{eff}<n_S$) the field assumes an unconventional oscillatory behavior, $E_x(y) = |E_0| \cos(\gamma_S y)$, and the maximum amplitude of the light is concentrated inside the middle slab.

In Fig. 1b we show the transverse wavevector of the propagating modes extracted from our simulation. One can see that the wavevector bifurcates and the transition, from purely imaginary (guided modes) to purely real (radiation modes), occurs exactly at the critical point where the wavevector vanishes ($k_S = 0$). This critical point occurs exactly when the refractive index of the middle layer ($n_S$) is equal to the effective index of the mode ($n_{eff}$) of a waveguide structure formed without the middle layer, i.e., $n_{eff} = n_S$, when $d = 0$. It also corresponds to the cut-off frequency of the asymmetric slab waveguides formed by half of the symmetric structure in Fig. 1a when the size of the middle slab tends to infinity, $d \to \infty$. In this case, when $k_S$ is real the field becomes radiative and therefore extremely lossy. In contrast, due to

the symmetry in the structure, the radiative field remains confined to the middle region, thus preventing radiation loss. The mirror-symmetric structure, as the one presented here, allows one to access these previously inaccessible modal regimes (see Supplementary Section 1). Furthermore, these results are complemented by a ray-optics picture presented in Supplementary Section 2. We employ ray-optics concepts to show that the critical point corresponds to the critical angle ($\theta_{inc} = \theta_{cr}$), considering the interfaces between the high-index and intermediary-index slabs ($n_H / n_S$), creating surface waves, while when the incidence angle is below the critical angle ($\theta_{inc}<\theta_{cr}$) and the refracted rays stay trapped in the middle slab by the outer interfaces, preventing radiation losses (see Supplementary Section 2).

We show the scale invariance of the optical mode at the critical point, when $k_S$ becomes zero, and the decay length of the evanescent field, which is defined as $\tau_S = 1/\gamma_S$, diverges to infinity and the field distribution becomes independent of the middle-slab's size (see Supplementary Section 1). In Fig. 1c we show the field amplitude as a function of the middle layer's thickness $d$. One can see that when $\gamma_S = 0$ and the symmetry is preserved, the thickness of the middle-slab material ($n_S$) can be scaled without affecting the properties of the characteristic mode. Note that the field can be confined to regions much larger than the ones given by the slot waveguide[18,19], which enables enhanced light-matter interaction (see Supplementary Section 3). Furthermore, at the critical point, the effective index of the propagating mode is independent of the thickness of the waveguide (see Fig. 1c). Also note that even for small variations around $\gamma_S = 0$ ($n_{eff} \cong n_S$), the field is expected to remain fairly independent of the geometry dimensions and wavelength. This invariance in the effective index and minimal dependence on the waveguide's thickness ensures the robustness of the propagating optical mode (see Supplementary Sections 4–9). In Supplementary Section 4, we show that this structure is also robust to optical losses. The independence of the materials' index contrast is discussed in Supplementary Section 6. In Supplementary Section 8 we show that, although the field distribution changes from a uniform distribution to a concave or convex distribution according to the wavelength (equivalent to those shown in Fig. 1b insets), the field overlap remains almost unchanged inside the middle layer. For a wavelength larger than the designed wavelength, i.e., $\lambda>\lambda_0$, even more light will be concentrated in the middle layer material. Besides that, even though the previous analysis is done for the TE polarization on a vertical structure, these effects are very general and

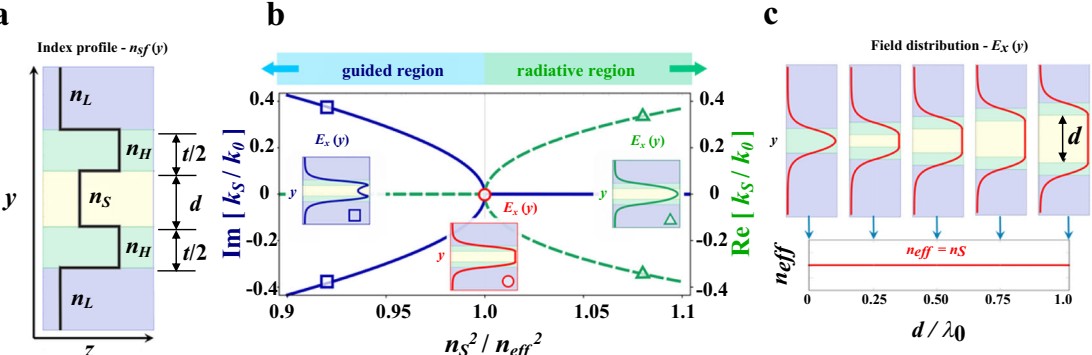

**Fig. 1 | Guiding light at the critical point and beyond. a** Refractive index profile of a symmetric slab waveguide structure (black line). The structure is formed by two high-index layers ($n_H$) of thickness $t/2$, separated by an intermediary-index middle layer ($n_S$) of thickness $d$, and cladded by a low-index material ($n_L$), where $n_H > n_S > n_L$. **b** Bifurcation diagram of the transverse wavevector at the critical point. The transverse wavevector inside the middle layer $k_S$ changes from purely imaginary (solid blue line) to purely real (dashed green line) as $n_S$ is increased. Correspondingly, the field profile in the middle region (inset) transitions from evanescent below the critical point (blue square), to uniform at criticality (red circle), and then

oscillatory above it (green triangle). This critical point occurs when the refractive index of the middle layer is equal to the effective index of the mode of the waveguide structure formed without the middle layer, i.e., $n_{eff} = n_S$, when $d = 0$, as a consequence of a vanishing wavevector ($k_S = 0$). **c** At the critical point the field profile remains uniform regardless of the middle layer's thickness ($d$). The field amplitude distribution is normalized by its maximum for each case. As the middle layer's thickness can be increased arbitrarily, this invariance allows for the field to be localized inside a region of lower refractive index ($n_S$).

can be applied to horizontal structures as well as for the TM polarization (see Supplementary Sections 1 and 5).

This uniform field distribution has also emerged in a quantum mechanical context, where the refractive index and electric field profiles are replaced by one-dimensional step potentials and wavefunctions[27–30]. In this context, it has been suggested that the potential landscape can be further engineered to create wavefunctions of arbitrary shape. By combining the insight gained from these quantum mechanical results with the integrability allowed by nanophotonic structures, such arbitrary potentials can be explored to create useful optical patterns. Besides this analogy, scale invariance can be also understood by invoking the notion of epsilon-near zero metamaterials[31], which also gives rise to fields with uniform phase distributions over extended regions[32,33]. Alternatively, the critical point can be analyzed as a square root branch point or singularity of the scattering matrix (S matrix)[34]. Furthermore, more recently, these cut-off points in dielectric waveguides were identified as special kinds of exceptional points[35], in which the system should lose dimensionality[35–37]. In this view, the scale invariance of the optical mode at the critical point can be seen as a direct physical manifestation of this effect.

The localization in the low-index material and scale invariance of the optical mode at the critical point also extends to 2D waveguides, referred to hereafter as *scale-invariant waveguides*. Figure 2a shows a 3D schematic of the scale-invariant waveguide with the electric field distribution profile of the propagating mode shown in the inset. One can see that our design exhibits a uniform field distribution in the vertical direction, with most of the mode confined in the lower-index material, in contrast to the Gaussian-like profile of the standard strip waveguide. We engineer the scale-invariant waveguide to operate at the critical point (at the critical angle) by dimensioning the high-index material layers such that the mode effective index matches the critical point at the telecom wavelength ($\lambda_0 = 1550$ nm). Note that, due to the vectorial nature of the 2D electromagnetic field, the critical point occurs for an effective index slightly below the middle-layer index ($n_{eff} < n_S$). We develop a simple analytical-based method to design it, which is presented in Supplementary Section 5. One can see in Fig. 2b that the critical point induces the extension of the optical mode irrespective of the thickness of the middle layer $d$. This scale-invariance

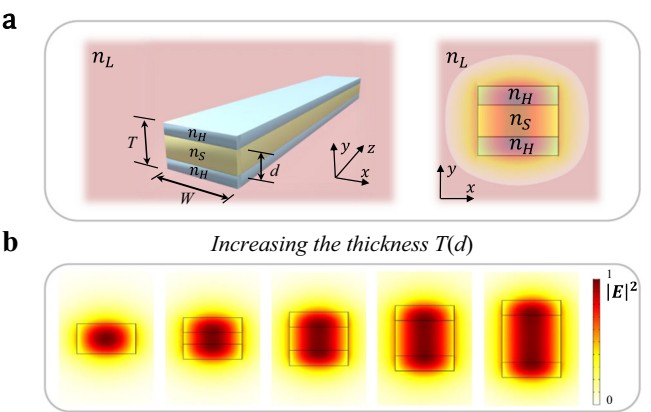

**a**

**b**

*Increasing the thickness T(d)*

**Fig. 2 | Unusual light properties of the scale-invariant waveguide at the critical point. a** 3D geometric schematics of the scale-invariant waveguide formed by a middle layer of the intermediary-index material ($n_S$), embedded between two layers of the higher-index material ($n_H$), surrounded by the lower-index material cladding ($n_L$), where $W$ and $T$ are the waveguide's width and total thickness, respectively. The scale-invariant waveguide shows a uniform field distribution in the vertical direction in contrast with the Gaussian-like distribution in the strip waveguide. **b** The electric field intensity distribution of the proposed waveguide as a function of the middle-layer's thickness $d$, showing the vertical scale invariance of the optical mode.

property can be tailored to control the amount of overlap with the lower-index material ($n_S$). We also note that these effects can be further extended to horizontal and circular structures (Supplementary Sections 5 and 6).

## Scale invariance of the optical mode

In order to experimentally verify guiding at the critical point (the critical angle), as well as the scale invariance of optical mode, we measure the effective refractive index for the scale-invariant waveguide with different sizes. We fabricate length-imbalance Mach-Zehnder interferometers (MZIs) composed of scale-invariant waveguides. The microscope image of MZI devices is shown in Fig. 3a. We fabricate the scale-invariant waveguides, with mirror symmetry in the vertical direction, by depositing a silicon oxynitride layer ($n_S = n_{SiON} = 1.75$) in between two silicon nitride SiN layers ($n_H = n_{SiN} = 1.99$), with a cladding of silicon dioxide ($n_L = n_{SiO2} = 1.44$), as shown in Fig. 3b. The width of the waveguide ($W$) is chosen to be 1 μm, while the thickness of the silicon nitride layer ($t_{SiN}/2$) is targeted to be 226 nm. The waveguide's total thickness is $T = d + t_{SiN}$, where $d$ is the thickness of the middle layer (see Fig. 2a and methods for fabrication details). The devices are engineered to guide light at the critical point for the TE fundamental mode at the center wavelength of 1.55 μm. We choose TE polarization for our experiments, in order to ensure no influence of the slot waveguide effect[18]. For comparison, we also fabricate and measure standard SiN strip waveguides ($n_S = n_H = n_{SiN}$), with the equivalent width $W$ and total thickness $T$, to contrast with the proposed device. Details of the design methodology are provided in Supplementary Section 7.

We demonstrate that the mode's effective index ($n_{eff}$) remains constant for different waveguide dimensions as predicted. In Fig. 3c, we show the measured normalized MZI transmission spectra (solid blue lines) and its fitting curve (dashed red lines). We extract the effective indices from the dispersion curves of the MZI transmission spectra[38] (see methods). Figure 3d shows the extracted effective indices of five different waveguides with the middle layer's thickness of $d = 0, 0.2, 0.4, 0.6, 0.8,$ and 1.0 μm, for the wavelength of 1.55 μm. For comparison, we also show the indices of the equivalent standard SiN/SiO2 waveguides (blue dots). The predicted values for the scale-invariant waveguide as well as for the standard strip waveguide, extracted from our FDTD (Finite-Difference Time-Domain) simulations, are also shown in dashed lines. The insets in Fig. 1e show the FDTD simulations of electric field intensities profiles, $|E|^2$, of the fundamental TE modes for each of the middle layer's thickness $d$. One can see that, in contrast to the standard waveguide, where the effective index is highly dependent on the waveguide's geometry, the effective index of the proposed waveguide stays almost constant. The results agree well with the predicted values and confirm the scale invariance of the mode profile. These measurements also show that for larger wavelengths (or thinner silicon nitride layers), light is still confined in the low-index material, as predicted (see Fig. 1b).

## Non-Gaussian field distribution

We image the near-field spatial profile of the optical mode in the scale-invariant waveguide and confirm the non-Gaussian uniform field distribution. In order to measure the mode transverse profiles, we use a knife-edge scan approach, assisted by a tapered nanofiber and nanopositioner[39]. We extract the mode profiles of the standard and scale-invariant waveguides by imaging the centers of the waveguides and using a deconvolution algorithm[39–41] (see Methods). Figure 4a, b shows the expected mode profile obtained from the FDTD simulation of the fundamental (TE) mode distribution of the standard SiN waveguide and the corresponding measured mode profile exhibiting similar Gaussian-like distributions. The expected and measured mode profile for the scale-invariant waveguide are shown in Fig. 4c, d. One can see that in strong contrast to the standard waveguide with its typical

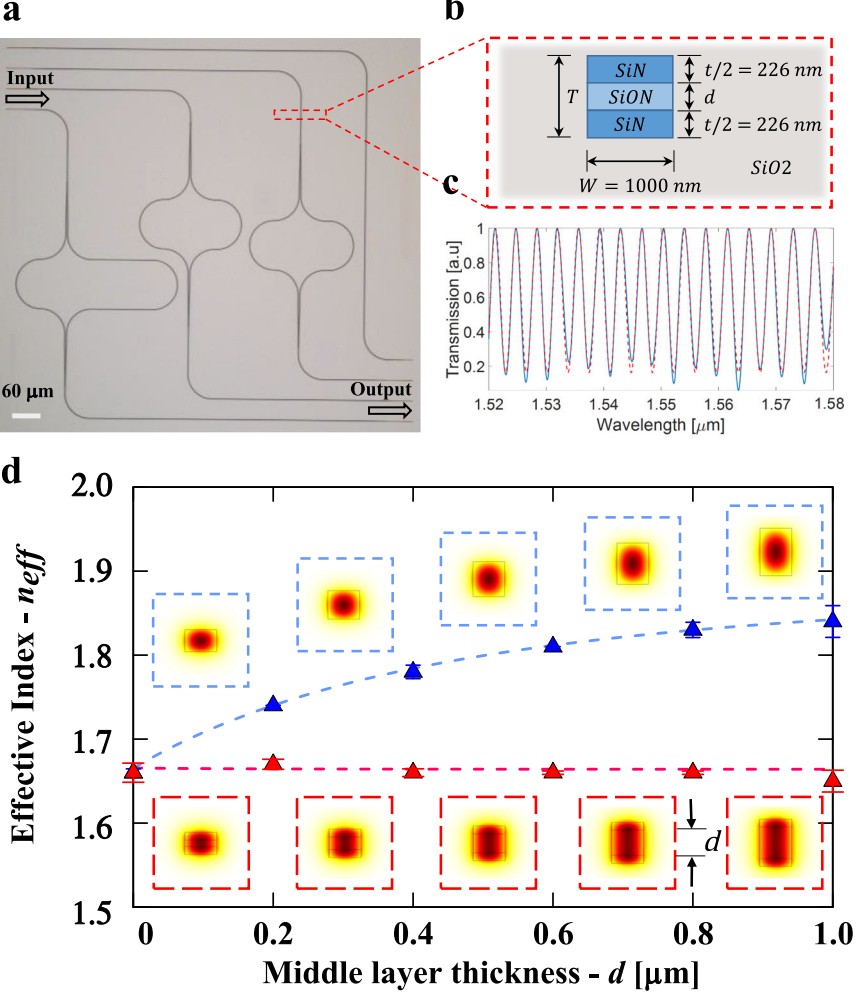

**Fig. 3 | Experimental demonstration of the scale invariance of the optical mode near the critical point. a** Optical microscope image of the three different-order low-pass MZI filters used to measure the dispersion curves of the scale-invariant waveguide and the strip waveguide. **b** Illustration showing the geometry stack of the scale-invariant waveguide formed by two SiN films ($n_H = n_{SiN} = 1.99$), separated by a SiON film ($n_S = n_{SiON} = 1.75$), and cladded by SiO2 ($n_L = n_{SiO2} = 1.44$). **c** Measured transmission spectrum of one of the MZIs' filters (solid blue line) and its fitting curve (dashed red line) as a function of the wavelength. Experimental and simulation values of **d** the effective index $n_{eff}$ as a function of thickness $d$ of the middle layer for the two waveguide configurations. The measured values are shown for the standard SiN strip waveguide (blue curve) ($1.666 \pm 0.001$, $1.741 \pm 0.09$, $1.782 \pm 0.012$, $1.812 \pm 0.002$, $1.830 \pm 0.011$, $1.842 \pm 0.023$) and for the scale-invariant waveguide (red curve) ($1.662 \pm 0.012$, $1.668 \pm 0.011$, $1.664 \pm 0.009$, $1.664 \pm 0.006$, $1.664 \pm 0.005$, $1.647 \pm 0.016$) The insets show the FDTD simulations of electric field intensity profiles, $|E|^2$, of the fundamental TE modes for each measured point at the wavelength of 1550 nm.

Gaussian-like profile, the scale-invariant waveguide exhibits an almost uniform field distribution, and most of the light is concentrated in the low-index material.

## Discussion

In summary, we have proposed and demonstrated an unconventional guiding mechanism in low-index materials. We showed that this mechanism exhibits unusual optical properties including a non-Gaussian field distribution and scale invariance of the optical mode. We designed and fabricated the scale-invariant waveguide in a CMOS-compatible SiN/SiON platform and experimentally demonstrated such properties, which can be explored to control and enhance light-matter interaction in low-index materials in heterogeneous waveguide structures. Although our demonstrations were done using a relatively high-index contrast, in Supplementary Section 6 we show that this effect is independent of the index contrast. For this study, we restricted the device dimensions to be single mode. However, these effects are also extendable to higher-order modes and multilayer systems, as described in Supplementary Section 1. Furthermore, these results can be also generalized for other structures, such as circular and subwavelength waveguides. In the Supplementary material, we show that the electric-field-intensity confinement factor in the low-index material is very robust against large variations in the wavelength, around the designed point, and against fabrication imperfections. Moreover, although our current demonstration is designed for operation in the telecommunication range, the same concept can be applied to other regions of the electromagnetic spectrum, for example, the visible or the terahertz range. We envision that the proposed structure will find applications in electro-optic modulation, nonlinear integrated photonics, and in temperature control. As another important application, our scale-invariant waveguide can be applied to manage saturation in the gain medium in semiconductor lasers.

## Methods

### Devices fabrication

The devices are fabricated on top of a nominal 4 μm-thick SiO₂ layer grown by thermal wet oxidation of 100 mm silicon wafer. An additional 2 μm-thick SiO₂ layer is deposited through plasma-enhanced chemical vapor deposition (PECVD) to ensure an index-profile symmetry with the top cladding. The thin-film layers of Si₃N₄ and SiON are deposited

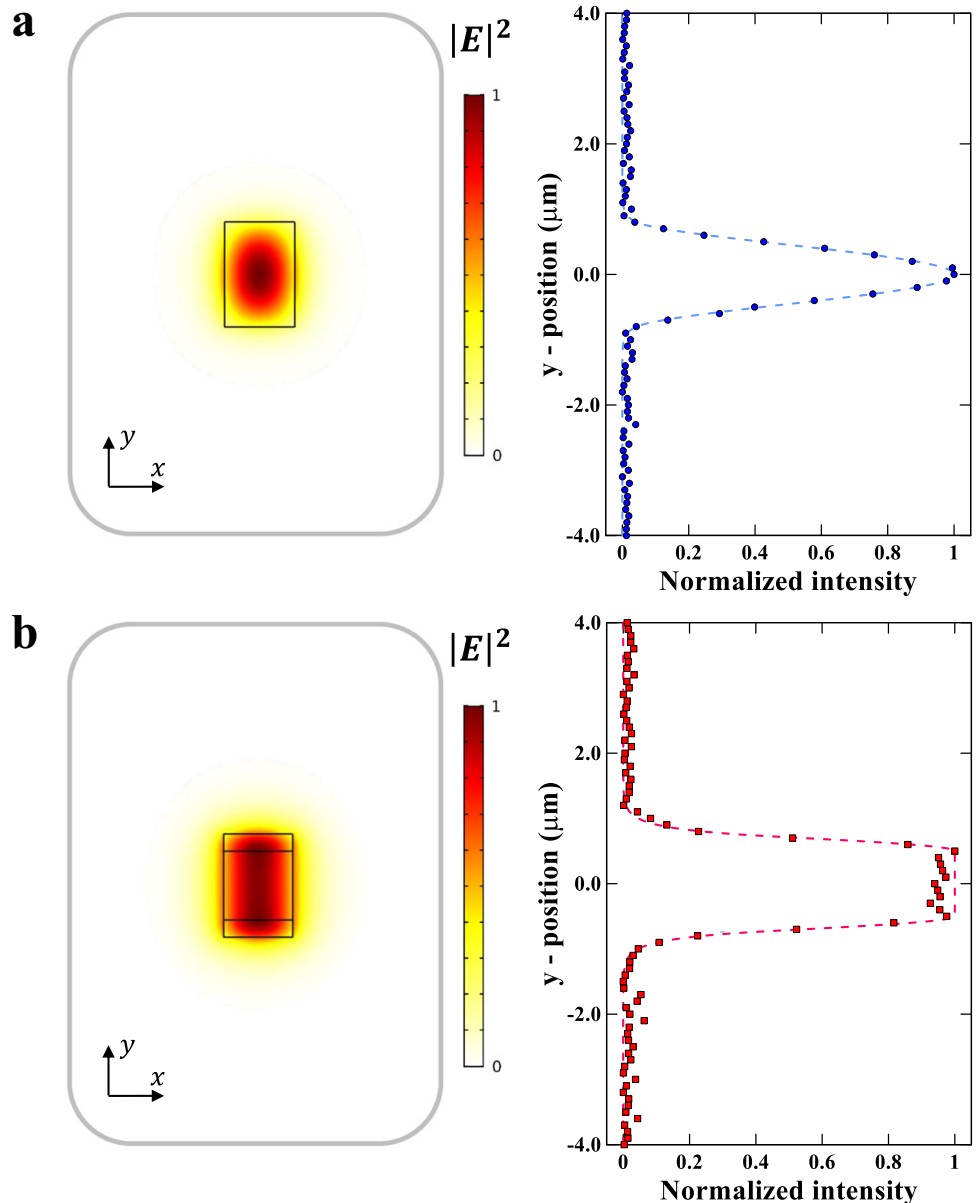

**Fig. 4 | Experimental demonstration of the non-Gaussian mode with a uniform electric field intensity near the critical point. a** FDTD simulation of electric field intensity profiles $|E|^2$ of the fundamental Transverse Electric (TE) mode profile of the strip SiN waveguide. **b** Measured mode profile of SiN strip waveguide showing a conventional Gaussian-like profile. **c** FDTD simulation of electric field intensity profiles $|E|^2$ of the fundamental Transverse Electric (TE) mode profile of the scale-invariant waveguide. **d** Measured mode profile of the scale-invariant waveguide showing an almost uniform intensity distribution in the middle region.

by PECVD. The different device thicknesses are achieved by carefully characterizing the film deposition rates under controlled deposition time durations. The total thickness is deposited in at least three steps in order to avoid mechanical stress on the films. The material refractive indices and absorption coefficients of each film were previously characterized, as a function of wavelength, by ellipsometry. These experimental data are then fitted using Sellmeier-type equations (Supplementary Section 7). The device patterns are created by patterning the negative-tone resist ma-N 2410 by a 100 KeV electron beam lithography tool. The patterns are then transferred to the SiN/SiON films via inductively coupled plasma (ICP) etching with $CHF_3/CF_4$ chemistry. In order to compensate for the undesired lateral etching, specific width biases are used during the CAD for different thicknesses. A top 6 μm-thick $SiO_2$ clad is deposited by using PECVD. The devices were then diced, and their input and output faces were polished for improvement of the nanotaper and lensed-fiber coupling.

### Effective index and group index measurements

We use the measured values of refractive indices and absorption coefficients from FDTD simulations to design three unbalanced MZIs. The simulated effective index of the mode and the path length difference $\Delta L$ between the two arms are used to establish the desired filter orders. The high-order filter ($m = 150$) is used to determine the group index, while the two other low-order filters ($m = 10$ and $m = 11$) are used to measure the effective refractive index[38]. We use a tunable laser and a high-speed photodetector to measure the transmission spectra. The transmission spectra are then fitted using a Least-square regression method in Matlab, like the one presented Fig. 3c. These steps are then repeated for each of the waveguide thicknesses previously described.

### Mode profiles

We first characterize the lensed-fiber mode spatial profile by using a conventional knife-edge experiment. For that, we use a thin metallic

razor blade mounted on a high-precision (20 nm) nanopositioner 3D system to control the translation of the blade. We measure the output signal by using a photodetector. After a position optimization in the $z$-direction to the focal point, the blade is translated in the $y$-direction and the output intensity is recorded. The measured data are then fitted by an error function and the corresponding Gaussian beam profile is recovered by its derivative. In the sequence, we replace the metallic blade with another lensed fiber. We maximize the output intensity in the $x$ and $z$ directions and translate the output fiber in the $y$-direction while recording the output intensity. We use a convolution algorithm implemented in Matlab to obtain the fiber profiles (see Supplementary Section 10). We compare both the results and observe quite a good agreement between them.

In order to recover the modal spatial profiles, we select the thickest SiON and SiN waveguides ($T = 1452$ nm) and remove the output nanotapers by mechanical polishing. We then use the nanopositioner system to control the position of the output fiber. The output fiber is placed as close as possible to the waveguide output by determining the optimal focal point in the $z$-direction, in order to avoid the output mode divergence and ensure a near-field measurement. The distance between the lensed fiber and the waveguide output is estimated using a visible camera to give a value of the order of 1 μm. By maximizing the output power, we determine the optimal horizontal position of the output fiber to match the center of the waveguide in the $x$-direction. Then again, we translate the output fiber in the $y$-direction and record the output. To overcome the ill-posed problem of the Gaussian deconvolution, we use a full-vectorial FDTD simulation input information and minimize the mean square error[39–42].

## Data availability

All data supporting the findings of this study are available from the corresponding author on request.

## Code availability

All code and simulation files not included in the paper or the supplementary information are available upon request from the corresponding authors.

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

## Acknowledgements

This work is supported as part of ARPA-E, PINE, Photonic Integrated Network Energy Efficient Datacenters program (DE-AR0000843); PIPES, Embedded Photonics ultra-bandwidth dense optical interconnect (EmPho) program (HR0011-19-2-0014); ARO, Novel Chip-Based Nonlinear Photonic Sources from the Visible to Mid-Infrared program (W911NF2110286); IMOD, Optimizing Microresonator's Based Sensor (OMA-1936345). National Science Foundation, All-photonic quantum network (OMA-1936345). Part of the fabrication of the devices was done at the Cornell NanoScale Facility, a member of the National Nanotechnology Coordinated Infrastructure (NNCI), which is supported by the National Science Foundation (Grant NNCI2025233); Columbia Nano-initiative; and the Nanofabrication Facility at the Advanced Science Research Center at The Graduate Center of the City University of New York. V.R.A acknowledges the Brazilian National Council for Scientific and Technological Development (CNPq) for research grants 310855/2016-0, 403031/2016-0, 403031/2019-2, and 306389/2021-5. We thank G. Bhatt, M. C. Zanarella, A. G. Molina, and M. C. Shin, for their collaboration and discussions.

## Author contributions

J.R.R., U.D., and M.L. conceived the project. J.R.R., A.A.G, R.G.J., and V.R.A. developed the models and the numerical simulations. J.R.R., A.M., X.J., I.D., E.S., and S.C. developed the fabrication process and fabricated the devices. J.R.R. and U.D. characterized the devices. M.L. supervised the project. All Authors contributed to the data analysis and writing of the manuscript.

## Competing interests

J.R.R. and M.L. are named inventors on US provisional patent application 17/119,328 regarding the technology reported in this article, and the rest of the authors of this work declare no other competing interests.
