## [Peer Review File · Nature Communications]

All-dielectric Scale Invariant WaveguideREVIEWER COMMENTS

Reviewer #1 (Remarks to the Author):

The authors have presented a new class of all-dielectric nanophotonic waveguide for the telecom range. Superficially, their structure appears to be an ordinary dielectric slot waveguide, and indeed, I was intending to raise this point in order to argue against the novelty of the work. However, the authors have clearly expounded the distinctions. Firstly, their new guide has a very strict design criterion: the effective modal index must be matched exactly to the bulk index of the sandwiched dielectric between the high-index layers, i.e. $n_S = n_{\text{eff}}$, meaning that the lateral decay constant becomes $\gamma_S = (n_S^2 - n_{\text{eff}}^2)^{1/2} = 0$, and so the decay function of the modal fields in the sandwiched region degenerates to a uniform distribution. As a consequence of this uniformity, the new class of waveguide is able to support much thicker low-index layers than a slot waveguide, thereby earning its "scale-invariant" namesake. Secondly, this technique can be applied to both polarizations, whilst a dielectric slot waveguides' mode only supports one.

This work is electromagnetically interesting, scientifically novel, well-written, and its conclusions are supported by experimental characterization. However, My sole criticism of this work pertains to insufficient information regarding the operation bandwidth over which the novel "scale-invariant" mode may be sustained.

The authors have highlighted restricted bandwidth as a key issue in other low-index guides, e.g. with statements such as "The few options available ... exhibit limited optical bandwidth ..." in the abstract, and "...guiding light in low-index materials has been achieved using ... However, all of these solutions are either lossy, require periodicity, exhibit narrow bandwidth, ..." in the second paragraph of the introduction. Despite this, there is no discussion whatsoever of the bandwidth of the scale-invariant nanophotonic waveguide. Given that all dielectric waveguides are dispersive (meaning that n_{eff} varies with respect to frequency), combined with the fact that n_S is a bulk-index (frequency-invariant), it is clear that the condition $n_S = n_{\text{eff}}$ is only met at one single frequency; a "critical point." On the other hand, the authors have claimed that "... even for small variations around $\gamma_S = 0$, the field is expected to remain fairly independent of geometry and wavelength." Well, how independent exactly? The authors must therefore provide a rigorous, detailed, and technical description of the variation in the guided mode with respect to frequency, with quantitative cut-offs that unambiguously define the span of frequencies over which this mode can be considered "scale invariant." It would be a good idea to include a new figure with field distributions that can visually show how the fields deviate from their desired behavior at the "critical point."

Reviewer #2 (Remarks to the Author):

In this manuscript, the authors propose and demonstrate the performance of an all-dielectric waveguide that shows no spatial invariance of the mode distribution. The structure is evaluated both numerically and experimentally showing agreement between the results. While the overall idea is interesting, it is unclear that the work will be of high significance in the field given that the presentation and analysis of some results need more work. Some theoretical statements that the authors use to support their results need further explanations. There are some comments that the authors may need to address for the manuscript to be suitable for publication in a journal. The comments are as follow:

1- In the abstract and the introduction, the authors mention "periodicity" as a bad aspect of some lower-index material-based waveguide designs. the authors should clarify what is the issue (if any)

with such structures and provide evidence by adding references. the way how this is supported in the manuscript is unclear as information of why there may be an issue is missing.

2- It is important to add references every time a statement is added (that is not supported by results). For instance, the text ending "... and mass adoption" in the "main" section first paragraph needs a reference. The end of page 2 also needs a reference to support the statement "...limits the amount of interaction". There are further instances where references are needed, the authors may want to re-read the manuscript and add them accordingly.

3- In page 4, it is mentioned "...this critical point occurs exactly when the refractive index of the middle layer.....". The authors should mention where this is shown. The same is valid for the cut-off frequency of the asymmetric slab in the same sentence.

4- In page 6, first paragraph. The first sentence is not supported by data. There is no figure (or at least there is nothing mentioned in the text that points to the figure) where the readers can see what happens when k_s becomes zero.

5- In supplementary material, figure S10, the axes (x,y,z) may be needed for the three panels.

6- Major comment: one of the novelties of the work, as mentioned in the manuscript, is the use of low-index materials. However, the structures reported in the paper still make use of high index materials. For instance, n_h in Figure 3 is 1.99 which is a relatively high value. There is a fine line between low and high index media (which is in fact subject to debate). The authors should clarify this aspect in the manuscript and support their statements with references.

7- Major comment: In the whole manuscript, the authors show the amplitude of the field distribution for the proposed scale-invariant waveguide and the results are compared with other types of dielectric waveguides (Fig. 1c, Fig. S1, S2...etc.). However, it seems like the normalized fields have been normalized to their own maxima not to a single reference (the case with $d=0$ for instance). This type of normalization will indeed create confusion as it may seem like the authors are using the same power as the input with the same maximum for all of them (a value of 1) which will be unphysical as the geometry of the waveguide is wider in the proposed case and the mode will distribute the power in the whole geometry. This should be corrected and a full study showing the limits of the proposed structure needs to be shown in terms of the decay of power at the middle layer when d is increased, for instance.

8- Major comments comments/results about losses in the dielectrics will be interesting to see.

9- Major comment: The authors suggest that there is no influence of d on the effective refractive index of the mode. By looking at the results from Fig. 3c, for instance:

- a. The contour plots need a scale bar and this needs to be renormalized to a reference not to each of the maxima for clarification, as mentioned in a comment above.
- b. The authors have added a straight line for the red triangles (experimental data of the proposed structure). However, there is some variations, and the line is not completely straight. Note that the authors have used a fitted line for the blue curve where the line (dashed blue) passed in the middle of each of the experimental triangular data points (blue triangles). This, however, is not the case for the red curve. The authors should mention and clarify this in the text.

10- Figure 4: the same comment about normalization applies here. The current way of showing the results needs to be updated and further discussed as mentioned above.

Reviewer #3 (Remarks to the Author):

The paper describes a waveguide structure composed of dielectric slabs of different refractive indexes, which are tuned to support a mode with a nearly uniform amplitude inside the central slab. The authors demonstrate in the supplementary material that this condition is achieved when the effective index of the guided mode is equal to the index of the central slab, and that it is independent of the slab thickness.

The analysis presented by the authors offers clear insights into the underlying physics of the waveguide structure, and extends the discussion by drawing analogies with other phenomena in different fields that share the same mathematical description. Furthermore, experimental findings are provided that demonstrate the nearly uniform mode inside the central slab, which stands in contrast to the typical Gaussian profile observed in slot waveguides.

While I appreciate the authors' efforts, I am hesitant to accept the paper for publication in a prestigious journal like Nature Communications for the following reasons:

1. The authors claim that the mode is confined inside the central slab, but this is not entirely accurate. To achieve matching, there must always be an evanescent field inside the material with index n_h , as seen in Figure 2(b). While the field is indeed nearly uniform and independent of the thickness inside the central slab, there must still be a field inside n_h to achieve matching.
2. The condition required to achieve this mode is a singularity that occurs only when $n_{\text{eff}} = n_s$. However, the paper does not discuss how the dispersion of materials can deviate from this condition and potentially compromise the performance of the structure.
3. Related to the previous point, there is no discussion in the paper about the operation bandwidth of the structure. It would be beneficial to address this and compare it with the bandwidth of a slot waveguide.
4. Another important consideration is that the paper does not address the transmission loss of the waveguide. While the modal profiles are shown throughout the paper, it would be valuable to know the propagation losses of the waveguide and compare them with those of a slot waveguide from a technological perspective.
5. Finally, it would be worthwhile to discuss the sensitivity of the mode to fabrication inaccuracies such as roughness and waveguide curvature.

I appreciate the authors' effort and the novelty of the proposed waveguide structure. However, based on the issues raised in my review, I cannot recommend the paper for publication in a leading journal such as Nature Communications.

Review Response for NCOMMS-23-13126

“All-dielectric Scale Invariant Waveguide”. We thank the reviewers for their insightful comments and favorable response to our manuscript.

The revised version of the manuscript has been modified to address the reviewers’ suggestions. We list here a summary of the changes to the manuscript and supplementary document based on the reviewers’ feedback in addition to the response to individual comments:

1. We have added discussions in the manuscript regarding the material losses, the usage of low-refractive-index materials, operation bandwidth, and the effect of fabrication imperfections, besides additional references, and directed the readers to the corresponding supplementary sections.
2. In supplementary section 4, we consider the effect of material losses as suggested by reviewers #2 and #3.
3. In supplementary section 6, we analyze the scale-invariant effect in low-index materials as suggested by reviewer #2.
4. In supplementary section 8, we discuss the effect of bandwidth on the waveguide and its relation with the scale-invariant effect as suggested by reviewers #1 and #3.
5. In supplementary section 9, we show the effect of fabrication imperfections on the proposed device as suggested by reviewer #3.

Reviewer #1 (Remarks to the Author):

The authors have presented a new class of all-dielectric nanophotonic waveguide for the telecom range. Superficially, their structure appears to be an ordinary dielectric slot waveguide, and indeed, I was intending to raise this point in order to argue against the novelty of the work. However, the authors have clearly expounded the distinctions. Firstly, their new guide has a very strict design criterion: the effective modal index must be matched exactly to the bulk index of the sandwiched dielectric between the high-index layers, i.e. $n_S = n_{eff}$, meaning that the lateral decay constant becomes $\gamma_S = (n_S^2 - n_{eff}^2)^{1/2} = 0$, and so the decay function of the modal fields in the sandwiched region degenerates to a uniform distribution. As a consequence of this uniformity, the new class of waveguide is able to support much thicker low-index layers than a slot waveguide, thereby earning its “scale-invariant” namesake. Secondly, this technique can be applied to both polarizations, whilst a dielectric slot waveguides’ mode only supports one. This work is electromagnetically interesting, scientifically novel, well-written, and its conclusions are supported by experimental characterization.

We thank the reviewer for their constructive comment on the main difference between the proposed device and the slot waveguide and regarding the novelty of our work. We would like to stress that although the demonstration was done in the telecom range this effect is very general and can also be used in other spectra range.

However, My sole criticism of this work pertains to insufficient information regarding the operation bandwidth over which the novel “scale-invariant” mode may be sustained.

The authors have highlighted restricted bandwidth as a key issue in other low-index guides, e.g. with statements such as “The few options available ... exhibit limited optical bandwidth ...” in the abstract, and “...guiding light in low-index materials has been achieved using ... However, all of these solutions are either lossy, require periodicity, exhibit narrow bandwidth, ...” in the second paragraph of the introduction. Despite this, there is no discussion whatsoever of the bandwidth of the scale-invariant nanophotonic waveguide. Given that all dielectric waveguides are dispersive (meaning that n_{eff} varies with respect to frequency), combined with the fact that n_S is a bulk-index (frequency-invariant), it is clear that the condition $n_S = n_{\text{eff}}$ is only met at one single frequency; a “critical point.” On the other hand, the authors have claimed that “... even for small variations around $\gamma_S = 0$, the field is expected to remain fairly independent of geometry and wavelength.” Well, how independent exactly? The authors must therefore provide a rigorous, detailed, and technical description of the variation in the guided mode with respect to frequency, with quantitative cut-offs that unambiguously define the span of frequencies over which this mode can be considered “scale invariant.” It would be a good idea to include a new figure with field distributions that can visually show how the fields deviate from their desired behavior at the “critical point.”

The reviewer is correct in stating that the scale-invariant effect only happens at a single frequency for a given structure. We carefully designed our device in order to demonstrate this effect exactly at 1550 nm wavelength. However, our claims regarding the broad bandwidth operation of the proposed device are related to the field overlap with the middle layer. Although the field distribution changes from a uniform distribution to a concave or convex distribution according to the wavelength, the field overlap remains almost unchanged over a bandwidth of 100 nm. We have added a new detailed text in supplementary section 8 discussing this point, including new figures with the field distribution deviations as suggested by the reviewer. We show that, once the device is designed to operate at the critical point, large variations in the wavelength around the designed point do not significantly alter the electric-field-intensity confinement factor inside the middle layer. For example, by considering a middle layer’s size of $d = 3\lambda_0$, which is impossible to conceive using a slot waveguide, the electric-field confinement factor is almost constant (with variation less than 2%) throughout the entire telecom band (C-band) and beyond, i.e., a 100 nm (~12.5 THz) bandwidth. We added new comments in this regard in the main text and directed the reader to the supplementary section for more details.

Reviewer #2 (Remarks to the Author):

In this manuscript, the authors propose and demonstrate the performance of an all-dielectric waveguide that shows no spatial invariance of the mode distribution. The structure is evaluated both numerically and

experimentally showing agreement between the results. While the overall idea is interesting, it is unclear that the work will be of high significance in the field given that the presentation and analysis of some results need more work. Some theoretical statements that the authors use to support their results need further explanations. There are some comments that the authors may need to address for the manuscript to be suitable for publication in a journal. The comments are as follow:

We appreciate the review for noticing the good agreement between our proposed structure and the obtained experimental results. We have added extra work as the reviewer has suggested.

1- In the abstract and the introduction, the authors mention "periodicity" as a bad aspect of some lower-index material-based waveguide designs. the authors should clarify what is the issue (if any) with such structures and provide evidence by adding references. the way how this is supported in the manuscript is unclear as information of why there may be an issue is missing.

Thanks to the reviewer for asking for further clarification on this. Our mention of periodicity is related to Photonic Crystal Waveguides (PCWs). In PCW, the cladding is composed of periodic structures, usually holes or pillars, which are placed around the low-index-material waveguide – considered a linear defect in the periodicity. The mix between holes (or pillars) with air creates a lower refractive index in the cladding than in the core, while its guiding mechanism is given by the mix of total internal refractions (TIR) and Brag reflections (due to the crystal band gap), which is dependent on the periodicity. Therefore, although a PCW can be used to guide light in low-index materials, its highly dependent on the periodicity which in turn increases the footprint of the device. In order to clarify this subject to the readers, we added more references about the PCWs and the mention of the device footprint required in the revised version of the manuscript.

2- It is important to add references every time a statement is added (that is not supported by results). For instance, the text ending "... and mass adoption" in the "main" section first paragraph needs a reference. The end of page 2 also needs a reference to support the statement "...limits the amount of interaction". There are further instances where references are needed, the authors may want to re-read the manuscript and add them accordingly.

We thank the reviewer for pointing this out, we have assumed that some of the topics are common knowledge. However, to avoid misinterpretation, we have added further references in all the sections mentioned.

3- In page 4, it is mentioned "...this critical point occurs exactly when the refractive index of the middle layer.....". The authors should mention where this is shown. The same is valid for the cut-off frequency of the asymmetric slab in the same sentence.

Thanks for noticing that, we refer to Fig. 1 in the revised version of the manuscript.

4- In page 6, first paragraph. The first sentence is not supported by data. There is no figure (or at least

there is nothing mentioned in the text that points to the figure) where the readers can see what happens when k_s becomes zero.

Thanks for pointing that out, we refer the reader to Fig. 1 and the experimental data in the revised version of the manuscript.

5- In supplementary material, figure S10, the axes (x,y,z) may be needed for the three panels.

We really appreciate the reviewer for pointing this out. We address this issue in the new version of the supplementary material.

6- Major comment: *one of the novelties of the work, as mentioned in the manuscript, is the use of low-index materials. However, the structures reported in the paper still make use of high index materials. For instance, n_H in Figure 3 is 1.99 which is a relatively high value. There is a fine line between low and high index media (which is in fact subject to debate). The authors should clarify this aspect in the manuscript and support their statements with references.*

Thanks to the reviewer for their observation. We agree that the terms high /low index media is relative. That being said, the results presented in the main manuscript, including the experimental data, were indeed obtained using a relatively high-index contrast between SiN ($n_H \approx 1.99$) and SiO₂ ($n_H \approx 1.45$): $\Delta = 23\%$. However, the proposed effect is very general and does not depend on specific values of the indices, other than the already mentioned relation $n_H > n_S > n_L$. To exemplify this point, we added a new supplementary material section 6, where we consider a waveguide formed with a lower index contrast ($\Delta = 1.36\%$), and designed a scale-invariant waveguide with a uniform profile. The results show the proposed effect is independent of the index contrast. Furthermore, we also show that, due to the low-index contrast, the Quasi-TE and Quasi-TM have almost the same field profile distributions. We added a comment about the independence of the index contrast in the main text and directed the reader to the supplementary section for more details.

7- Major comment: *In the whole manuscript, the authors show the amplitude of the field distribution for the proposed scale-invariant waveguide and the results are compared with other types of dielectric waveguides (Fig. 1c, Fig. S1, S2...etc.). However, it seems like the normalized fields have been normalized to their own maxima not to a single reference (the case with $d=0$ for instance). This type of normalization will indeed create confusion as it may seem like the authors are using the same power as the input with the same maximum for all of them (a value of 1) which will be unphysical as the geometry of the waveguide is wider in the proposed case and the mode will distribute the power in the whole geometry. This should be corrected and a full study showing the limits of the proposed structure needs to be shown in terms of the decay of power at the middle layer when d is increased, for instance.*

Thanks for the reviewer's comment. However, we firmly believe that presenting the mode electric field (or electric intensity) distributions normalized by their maximum value is not only a normal practice but also the most appropriate one. The reason is that the amount of power in each waveguide (or different cross-sections) depends on power coupling efficiency from the light source. In our experiment, each waveguide cross-section is a different device, which is composed of different middle layer thicknesses, as discussed in the materials and methods sections in the main manuscript. Therefore, we could not ensure that the same amount of power is coupled in all devices. The reviewer's comments regard to optical power normalization would totally make sense in the case of a unique device, in which the dimensions are varied along its length, which although perfectly possible, has not been addressed in this work. In fact, the electric field intensity at a specific transversal position would decay for a larger device width, considering the same optical coupled power. The transversal electric field amplitude provides a very important piece of information on the overall impact on the device performance for different parameters (dimensions and wavelength); however, it needs to be complemented by other performance metrics, such as the electric field intensity confinement factor, which we show for several cases, showing low impact of these parameters on the device performance.

8- Major comments /results about losses in the dielectrics will be interesting to see.

We agree with the reviewer, and following his suggestion, we have added a detailed text about the effect of material losses in supplementary section 4. We show that the proposed structure is robust against material losses for nominal loss values of most of the dielectric materials used in the Near-IR regime. For instance, we show that for material losses $\alpha \leq 10^{-2}$, the mode profile and its confinement factor in the middle layer stay constant even though its propagation loss increases. This level of loss includes the case for most dielectric materials currently in use in integrated photonics, where the loss values encountered in most of the foundries are well below 10 dB/cm ($\alpha < 3 \times 10^{-5}$). Furthermore, we show how the effect can also be explored to leverage light confinement even with materials with high levels of gain/loss as in laser media or metallic materials. We added a comment relayed to material losses in the main text and directed the reader to the supplementary section for more details.

9- Major comment: The authors suggest that there is no influence of d on the effective refractive index of the mode. By looking at the results from Fig. 3c, for instance:

a. The contour plots need a scale bar and this needs to be renormalized to a reference not to each of the maxima for clarification, as mentioned in a comment above. b. The authors have added a straight line for the red triangles (experimental data of the proposed structure). However, there is some variations, and the line is not completely straight. Note that the authors have used a fitted line for the blue curve where the line (dashed blue) passed in the middle of each of the experimental triangular data points (blue triangles). This, however, is not the case for the red curve. The authors should mention and clarify this in the text.

We appreciate the reviewers' comments regarding the normalization and variations in our experimental data. Please, see our justification in question 8, regarding the normalization question. Regarding our experimental data, please note that throughout our manuscript, we have tried to be very careful by not claiming that the experimental results were obtained exactly at the critical point. This is the reason why we have used sentences as *almost constant* effective refractive index and *almost uniform* field distributions. As discussed in the new sections included in the supplementary materials, wavelength variations, and fabrication imperfections would not allow us to make such a claim. However, we believe that our experimental data strongly support our theoretical predicted models. Furthermore, for the applications that we envisioned, the enhancement of the field overlaps with low-refractive-index materials, being exactly at the critical point is not strictly necessary.

10- Figure 4: the same comment about normalization applies here. The current way of showing the results needs to be updated and further discussed as mentioned above.

Please see our justification in question 8 regarding the normalization question.

Reviewer #3 (Remarks to the Author):

The paper describes a waveguide structure composed of dielectric slabs of different refractive indexes, which are tuned to support a mode with a nearly uniform amplitude inside the central slab. The authors demonstrate in the supplementary material that this condition is achieved when the effective index of the guided mode is equal to the index of the central slab, and that it is independent of the slab thickness.

The analysis presented by the authors offers clear insights into the underlying physics of the waveguide structure, and extends the discussion by drawing analogies with other phenomena in different fields that share the same mathematical description. Furthermore, experimental findings are provided that demonstrate the nearly uniform mode inside the central slab, which stands in contrast to the typical Gaussian profile observed in slot waveguides.

We appreciate the reviewer for their comments on the physics of the demonstrated effect and its connection with other phenomena, including in other areas.

1. *The authors claim that the mode is confined inside the central slab, but this is not entirely accurate. To achieve matching, there must always be an evanescent field inside the material with index n_h , as seen in Figure 2(b). While the field is indeed nearly uniform and independent of the thickness inside the central slab, there must still be a field inside n_h to achieve matching.*

We agree with the reviewer that, in order to achieve the matching, part of the mode distribution must still be inside the high-index layer (n_H) and also outside in the low-index cladding (n_L). We

claim that *almost* all light can be concentrated in the middle layer. This can be seen in Fig. S6(a), where we show that more than 95% of the light can be concentrated inside the middle layer. Although, we are aware that the confinement factor will never reach 100%, this high confinement factor is very impressive by itself. Even more, when compared with a conventional slot waveguide, made of the same materials, the maximum confinement that can be achieved is around 30%. In order to make sure this point is clear to the readers, we are going to add a comment on the main manuscript in the referred supplementary section.

2. The condition required to achieve this mode is a singularity that occurs only when $n_{eff} = n_s$. However, the paper does not discuss how the dispersion of materials can deviate from this condition and potentially compromise the performance of the structure.

In our design, the dispersion of the materials was taken into account by the previous characterization of all materials. In the new supplementary section 8, about the operation bandwidth, the dispersion of the material is taken into account.

3. Related to the previous point, there is no discussion in the paper about the operation bandwidth of the structure. It would be beneficial to address this and compare it with the bandwidth of a slot waveguide.

We appreciate the reviewer's insight into the device operation bandwidth. In order to not be repetitive, we kindly direct the reviewer to the answer that has been given to reviewer #1 on exactly the same question.

4. Another important consideration is that the paper does not address the transmission loss of the waveguide. While the modal profiles are shown throughout the paper, it would be valuable to know the propagation losses of the waveguide and compare them with those of a slot waveguide from a technological perspective.

We thank the reviewer for bringing up the topic of the effect of propagation loss of the waveguide. In order to not be repetitive, we kindly direct the reviewer to the answer that has been given to reviewer #2, item 8, on exactly the same question.

5. Finally, it would be worthwhile to discuss the sensitivity of the mode to fabrication inaccuracies such as roughness and waveguide curvature.

As per the reviewers' suggestion, we have included an extra section in the supplementary, section 9, that shows the effect of fabrication imperfections on the waveguide and on the waveguide curvature. For vertically-oriented stacks, the variations are less perceptible since in most deposition techniques the film thickness can be well controlled (down to a few nanometers), through careful

pre-characterization of their deposition rates. On the other hand, for horizontally-oriented stacks, there are differences between the layout mask's nominal widths and the fabricated dimensions. We show that, although this difference can be enough to take the scale-invariant waveguide out of the critical point, it does not significantly affect the electric-field-intensity concentration inside the middle layer. Furthermore, we show that a symmetric variation creates the same effect as the previously described wavelength variation, while an asymmetric variation creates an inclination in field distribution. We show that in all the cases, the confinement factor stays practically constant showing the robustness of the proposed structure.

Final comments and remarks.

In this manuscript, we proposed a pure dielectric device that guides light at the critical point (or the critical angle). We successfully demonstrated such behavior experimentally, for the first time to the best of our knowledge. The results show a good agreement with analytical models and rigorous FDTD simulations. We highlighted the connection with other areas of physics, including its intrinsic connection with the quantum behavior of quantum wells and non-Hermitian physics of exception points. Besides the demonstration of such an interesting physical effect, we prove that such a structure can be used to concentrate light in low-index materials, showing superior performance than the currently most used structure for such application, namely the conventional slot waveguide, which was previously proposed by our group. We show that, although the scale-invariant effect only happens for a unique set of parameters (wavelength, materials, and dimensions), the overlap of the electric-field intensity with the middle layer material is very robust to fabrication imperfections and material losses, as well as being spectrally broadband, which makes this waveguide structure very attractive to integrated photonics applications. Furthermore, we stress that although we demonstrated this effect in the telecom range, it was rigorously derived from Maxwell's equations and boundary conditions, without any assumptions of the spectral range; therefore, it may find applications also in other regions of the electromagnetic spectrum. Finally, we believe that with the helpful comments and insights of the reviewers and the editor, the revised version of the manuscript and its supplementary material are now suitable to be published in the renowned Nature Communications.

REVIEWERS' COMMENTS

Reviewer #1 (Remarks to the Author):

My sole query with regards to operation bandwidth has been addressed to my satisfaction and I endorse this manuscript for publication.

Reviewer #2 (Remarks to the Author):

The authors have addressed the comments from this reviewer. The authors have decided not to renormalize the results as suggested by reviewer 2. This reviewer considers that renormalization would make a comparison of results fair due to manuscript claim of being invariant and scalable. If the authors decide to keep it as it is, it will be important to add a comment in the caption stating the reason why the results have been normalized to their own maximum. Apart from this, this reviewer has no further comments.

Reviewer #3 (Remarks to the Author):

The authors have correctly addressed all my previous comments. Nevertheless, I have a final suggestion. In my opinion, the study of the bandwidth should be included in the main text. Perhaps, Fig. S15 and the accompanying text.

Review Response for NCOMMS-23-13126

“All-dielectric Scale Invariant Waveguide”. We thank the reviewers for their insightful comments and favorable responses to our manuscript.

The revised version of the manuscript has been modified to address the reviewers’ suggestions. We list here a summary of the changes to the manuscript based on the reviewers’ feedback in addition to the response to individual comments:

We have included comments about the field normalization and regard to the bandwidth study.

Reviewer #1 (Remarks to the Author):

My sole query with regards to operation bandwidth has been addressed to my satisfaction and I endorse this manuscript for publication.

We thank the reviewer for their time and constructive comments.

Reviewer #2 (Remarks to the Author):

The authors have addressed the comments from this reviewer. The authors have decided not to renormalize the results as suggested by reviewer 2. This reviewer considers that renormalization would make a comparison of results fair due to manuscript claim of being invariant and scalable. If the authors decide to keep it as it is, it will be important to add a comment in the caption stating the reason why the results have been normalized to their own maximum. Apart from this, this reviewer has no further comments.

As the reviewer has suggested, we have added a comment in the figure caption related to the normalization of field distribution to its maximum for each case. We thank the reviewer for their time and comments.

Reviewer #3 (Remarks to the Author):

The authors have correctly addressed all my previous comments. Nevertheless, I have a final suggestion. In my opinion, the study of the bandwidth should be included in the main text. Perhaps, Fig. S15 and the accompanying text.

As per the reviewers’ suggestion, we have included an extra sentence in the main manuscript explaining the changes in the mode profile as a function of the wavelength. We also directed the reader to the field profiles shown in the insets of Fig. 1(b) in the main manuscript and to the supplementary section 8 for more details. We thank the reviewer for their time and comments.

Final comments and remarks.

We would like to thank and recognize all the reviewers and editors for their constructive comments regarding our manuscript and its supplementary material. We firmly believe that this process has improved the clarity of our manuscript.